# Exploring Effective Conservation of Charismatic Flora: Orchids in Armenia as a Case Study

**Aisyah Faruk [1,\*]**, **Astghik Papikyan [2,3]** and **Anush Nersesyan [2,3]**

1   Royal Botanic Gardens Kew, Millennium Seed Bank Building, Ardingly RH17 6TN, UK
2   A. Takhtajyan Institute of Botany of National Academy of Science of Armenia, Acharyan 1,
    Yerevan 0040, Armenia; papikyanastghik@gmail.com (A.P.); annersesyan1@gmail.com (A.N.)
3   Nature Heritage NGO of the Republic of Armenia, Yerevan 0040, Armenia
\*   Correspondence: a.faruk@kew.org

**Abstract:** Plants are the bedrock of life on Earth but are increasingly threatened with extinction. The most cost-effective way of conserving plant diversity is through Protected Areas (PAs). However, the locality, size, and management of PAs are crucial for effectively maintaining diversity and have been criticized as currently inadequate. Using Armenia as our study site and orchids as our study taxa, we sought to (1) identify spatial patterns of orchid diversity hotspots and corresponding PA network sites; (2) examine if the current PA network is effective at capturing orchid species richness and diversity and (3) explore the relationship between the range of area suitability of species and level of protection. We used data collected from herbarium, field visits and GBIF occurrence records. Using freely available mapping software, we created heatmaps of observations and species richness. We compared PA sites based on the number of species (species richness) and diversity (Shannon–Weiner Index). Species range was developed using the MaxEnt model and a correlation analysis was performed against the proportion of the range within PA. We found that 57% of PA sites have a representation of at least one species of orchid, but some threatened species are not presented within any PA site. The Tavush and Syunik province not only held the highest species richness (>10 species), but the PA network within also held high orchid diversity (2.5 diversity index value for Dilijan National Park). We did not find a significant relationship between the range of area suitability for orchids and protection; however, all our target species had less than 30% of their range under protection. Our study highlights important challenges related to the limitations of available data, and we discuss these implications towards effective conservation outcomes for orchids for the region.

**Keywords:** biodiversity; Caucasus; endemics; exceptional species; important plant areas; KBAs; MaxEnt; Protected Area; QGIS; Species Distribution Models (SDMs)

## 1. Introduction

The urgency of protecting our world's biodiversity is now becoming apparent, with various landmark reports attracting public attention [1,2]. Key to this is the growing understanding that biodiversity is fundamental to all life on Earth and that anthropogenic impacts are the largest contributor to its decline [1–4]. The impact of our economic growth within the last 50 years has certainly contributed towards an estimated 1 million of all described species being threatened with extinction [2]. Plant and fungal diversity are the bedrock of healthy ecosystems and are worryingly showing the same downward trend across different continents [5–7]; globally, 39% of all vascular plants are now threatened with extinction [8].

Arguably, the best way of curbing biodiversity decline and protecting wild species is the establishment of Protected Areas (PAs), as these provide a safe refuge for both plants and animals within their natural habitats [9,10]. International biodiversity initiatives have consistently featured various targets that relate to a percentage of land designated for protection (e.g., Aichi Biodiversity Targets [11]). Although nations have met, or, in

some cases, exceeded these targets, the continued decline in biodiversity over the past few decades has cast some skepticism over their overall effectiveness [12–15]. Historically, the reasoning behind the establishment of PAs focused on areas that were remote or "wild", their aesthetics and/or sole use of natural resources [16]. As a result, many PAs were either not adequately positioned to capture species location [12,16,17], conserving genetic diversity [18] and/or ineffectively managed [19]. In recent years, the establishment of PAs has become more strategic—for example, the Natura 2000 sites in Europe [20]. Additionally, the use of tools such as Species Distribution Models (SDMs) is starting to feature in mainstream decision-making processes, ensuring that the development of PAs is based on ecologically sound evidence [21]. Although the use of more strategic and scientifically robust methods in determining PAs is welcomed, there is a bias towards areas important for megafaunas and "umbrella species", which can result in more cryptic species being left behind [16]. Additionally, there is increasing awareness of the phenomenon known as "plant blindness", where plants are generally overlooked by policy-makers, conservationists and the general public [22]. As with other less well-known or well-loved species, the limited or lack of attention given to plants in conservation action can exacerbate their rate of decline, thereby risking further ecosystem collapse [22].

Some of the most charismatic taxa within the plant kingdom are those within the family Orchidaceae, the second most diverse angiosperm plant family after Asteraceae, with an estimated 28,484 species worldwide [23]. The popularity in the study of orchids, both by academics and amateurs, could be attributed to various reasons, from simply the beauty of its flowers, the curious co-evolution with pollinator systems [24–26], the remarkable cases of sexual deception in some genera such as *Ophrys* L. [27,28] and/or the complex and, at times, very specific ecology and relationship with mycorrhizal fungi [29,30]. It is their complex ecological specialization that makes orchids an ideal taxon for testing the effectiveness of various conservation strategies, as they can be used as bioindicators of ecosystem health [31].

In Europe, all orchids are of the terrestrial form, with the mountainous regions of modern Europe, Asia Minor, Middle and Central Asia and the Caucasus recognized as important areas for some taxa (e.g., *Epipactis* and *Dactylorhiza*) [32]. Despite its popularity, there are major gaps in knowledge for orchids, particularly in parts of West Asia [33], leaving species within these biodiversity-rich areas exposed to increasing threats of extinction. Although some species are found to have a broad range, stretching across Europe (e.g., *Anacamptis pyramidalis* (L.) Rich. [34]), others have evolved such specialist relationships with their immediate environment, restricting their range to only a small area (e.g., *Goodyera macrophylla* Lowe [35]). With increasing levels of habitat loss, fragmentation and the ongoing effects of a changing climate, terrestrial orchids with narrow ranges are likely to experience a greater risk of extinction in the coming decades [31,36]. Therefore, a substantial proportion of these narrow endemics will need to be accounted for within PA networks to enable the conservation of the entire species to be effective.

The Caucasus Biodiversity Hotspot holds c. 7000 species of vascular plants, of which 25% are classified as endemic to the region [37]. Within this biodiverse region is Armenia, a small landlocked country within the South Caucasus. Although the country is relatively small (territorial area of 29,743 km$^2$), more than 3800 vascular plant species are found within, making up more than a half of the Caucasus flora. Additionally, there are elements of both horizontal and vertical habitat level zoning found here, resulting in a complex landscape structure with a wide range of different habitat types, from semi-desert zones to alpine grasslands and moist temperate forests. Therefore, we would expect that the PAs within Armenia would encapsulate a wider range of different bioclimatic and ecological types across its network. In the last decade, there has been a significant expansion of PA networks across the Caucasus, with Armenia incorporating Lake Arpi National Park [38], Arevik National Park [39], Khustup Sanctuary [40] and the Zangezur Sanctuary [40] within its network. The current PA network within Armenia consists of 34 sites, the majority of which are recognized as nationally designated Protected Areas within the IUCN Category

IV (27 sites), four under IUCN Category II and three under IUCN Category 1a. There are also two internationally recognized Ramsar sites, namely Lake Arpi to the north of the country and the Khor Virap marshlands within the Ararat Province [41]. Although the continued expansion of the protected network is welcomed, there is an acknowledgement that gaps are present. Indeed, the new PAs were incorporated with the aim of enhancing the protection of the endangered Persian leopard, and there has been little attention towards key plant groups or families within the prioritization process [37].

In this paper, we aim to explore whether the current establishment of PAs is effective at conserving orchid diversity. We highlight hotspots for orchids and identify the richness and diversity within current PA networks. We hypothesize that orchid species suited to thriving within a narrow range of areas, i.e., higher extinction risk, will be underrepresented within the current PA network and therefore should be prioritized for further conservation action.

## 2. Materials and Methods

### 2.1. Data Collection and Cleaning

Data on species name, locality (GPS, locality notes and/or descriptions), collection dates, altitude and, where possible, collector and verifier were gathered from herbarium specimens of orchids deposited in the Herbarium of the A. Takhtajyan Institute of Botany of NAS RA (ERE) and the Komarov Botanical Institute of the Russian Academy of Sciences, St. Petersburg Herbarium (LE) (herbaria acronyms according to [42]). We used the QGIS GBIF plugin [43] to download data on orchids from the Global Biodiversity Information Facility (GBIF) occurrence database [44]. We used data (taxonomy, locality, altitude and data source) derived from preserved specimens (i.e., herbarium vouchers) and human observations. Species that were found only within the GBIF human observation dataset were removed to reduce the possibility of misidentification. Additional fieldwork between the months of May and June 2020 was conducted to confirm orchid locality and identify new populations. Ecology and flowering times were obtained through herbarium studies and the published literature [45,46]. The nomenclature of the resulting species list of orchids was validated through name matching using the World Checklist of Selected Plant Families (WCSP) [23] and the World Checklist of Vascular Plants (WCVP) [47].

For herbarium collections, labels of 931 herbarium specimens, representing 39 species and 16 genera, were initially transcribed, and by using the published literature on the habitat preferences of each species, and location description, geographic coordinates for each specimen were obtained through Google Earth (version 2020). A total of 58 vouchers that were collected between 1929 and 1980 were omitted from the subsequent analysis as their locality could not be verified. For field observations, we omitted records where the field observers could not readily identify the plant to species due to lack of flowers during the time of the visit. Finally, for the GBIF records, we limited our investigation to those identified to species.

The Protected Areas (PAs) shapefile was downloaded from the World Database of Protected Areas (WDPA) through the Protected Planet website [41] and cropped for the Armenia territory shapefile obtained from the freely available resource, the DIVA-GIS website [48].

### 2.2. Sampling Density and Species Richness Analysis

We used a freely available mapping software, QGIS (version 3.10.12-A Coruña), to project the 1101 geocoordinates from the resulting orchid dataset. A heatmap of the collection data was produced using the Heatmap Tool to identify the pattern of orchid distribution according to presence-only data. The tool uses Kernel Density Estimation with a radial distance buffer of 5 km. Where 10 or more points and their radial buffer intersect, these would be identified as a cluster. We overlaid the PA network shapefile with the point coordinates of orchid species and, using the "point sampling" tool in QGIS, extracted orchid occurrence points within each PA site to identify which PA was better at

capturing orchid species diversity using the Shannon–Weiner diversity index found within the "vegan" package [49] in R software 4.0.2 [50].

We identified areas of high species richness across the study area by overlaying 5 km × 5 km grids across the study area and using a readily available algorithm within the QGIS toolbox ("Count points in polygon"), and we counted the number of unique species within each grid cell. We specified the accepted taxonomic names for each species as the unique class field.

### 2.3. Species Distribution Models (SDMs) of Conservation Priority Species

We used Species Distribution Models (SDMs) to develop the potential areas suitable for orchid species found in Armenia. We used a maximum entropy (MaxEnt) approach to model orchid species found within our study system using the MAXENT software (version 3.4.3) [51], which is the preferred method for presence-only data (e.g., herbarium specimens, etc.), and handles small datasets relatively well [52]. SDMs were done for species that had 10 occurrence data or over.

Previous studies have found various factors that can influence orchid distribution across a landscape, including pollination specificity and mycorrhizal association [31]. As both factors are also found to correlate with altitude [53–56], we used this as proxy variables within the SDMs. Alongside altitudinal data, we also retrieved climatic variables relating to precipitation and temperature from the World Clim dataset (worldclim.org, 2020). These were clipped to the Armenia territory outline and bioclimatic variables from this monthly climate data were extracted using the *biovars* function within the "dismo" package [57] within R statistical software [50]. Environmental variables of orchid localities were extracted using the "Point Sampling Tool" on QGIS. A Principal Component Analysis (PCA) was used to identify correlations across variables using R software. To reduce the risk of overfitting our models, we took the variables strongly associated with the first four Principal Component axes as our environmental variables for the subsequent models.

The MAXENT software generated 10,000 random points within our study area as "absence" points and developed predicted distribution using the Jacknifing method. For model validation, we used 25% of the presence-only data and *cloglog* as the output format, as recommended by [58]. We used linear, quadratic and hinge features to run the models using the auto feature turned on within the MAXENT software. To identify the model with the most appropriate regularization parameter, we ran models with regularization multiplier values from 1 to 10 for each species and compared the models using the Akaike's information criterion [59] corrected for small sample size (AICc), following the methodology based on Warren and Seifert [60]. The model with the lowest AICc value was taken as the one to move forward with in subsequent analysis as this represented the model that was the most parsimonious.

We calculated area suitability ranges with high predicted conditions by converting the output raster with prediction values above 0.7 into a vector and calculating this area using the "Add vector geometry" tool in QGIS, which gives the area in km$^2$ per pixel. The mean range size for each species within Armenia was calculated. An overlap analysis was done with the PA shapefile layer to extract the percentage of potential species distribution within PA networks. A correlation analysis between mean potential species range and mean percentage of range protected was done using R software, and Pearson's correlation coefficient was calculated to test the null hypothesis outlined in the Introduction.

### 3. Results

#### 3.1. Sampling Density and Overall Orchid Diversity

The oldest specimen found within the ERE herbarium was collected in 1916 by B. Shishkin and N. Abzianidze and identified as *Orchis palustris* Jacq., later recognized as a synonym of *Anacamptis palustris* (Jacq.) R. M. Bateman, Pridgeon & M. W. Chase [23]. The most recent herbarium specimen used for the study was *Anacamptis morio* (L.) R. M. Bateman, Pridgeon & M. W. Chase (syn *Orchis morio* (L.)), collected in 2017 by A. Nersesyan,

A. Papikyan, S. Galstyan and N. Hayrapetyan near Goris, a town in the southeast of Armenia. Based on herbarium vouchers and the published literature, the earliest flowering orchid was *A. pyramidalis* (April), while the latest flowering orchids were *Epipactis persica* (Soó) Hausskn. ex Nannf. and *Epipogium aphyllum* Sw. (August). Generally, orchids found in Armenia tended to bloom between May and July [45].

After data cleaning and accounting for synonyms, an initial list of 48 species was reduced to a total of 43 accepted species and subspecies from 14 genera of orchids found in Armenia, with 18 identified as synonyms according to the WCSP and WCVP (Supplementary Materials Table S1 Orchid species occurrence data). *Dactylorhiza maculata subsp. fuchsii* (Druce) Hyl., *D. maculata* (L.) Soó. and *Orchis* x *calliantha* Renz & Taubenheim records were omitted from subsequent analysis as these were the only species found in the GBIF human observation dataset and not verified elsewhere. There were 1101 occurrence data captured from the resulting list of species from the various sources. We found 17 clusters, where 10 or more points and their 5 km radial buffer overlapped. The majority were found to the south of Armenia within Syunik province, which also had the largest cluster, followed by Tavush province, towards the north-east of the country (Figure 1a). We also found a small cluster within Lori province, close to the Gyulagarak PA site, and in the northern part of Sevan Lake, within Gegharkunik province. A similar pattern was found in relation to species richness, with Syunik having the highest number of species (16) within a 5 km-by-5-km grid, followed by Tavush province with 12 species and the site in Lori with eight species within a 5 km-by-5-km grid square (Figure 1b). In the Ararat and northern Kotayk provinces, we found small observational clusters, but with relatively low species numbers (2 to 5 species) (Figure 1a,b).

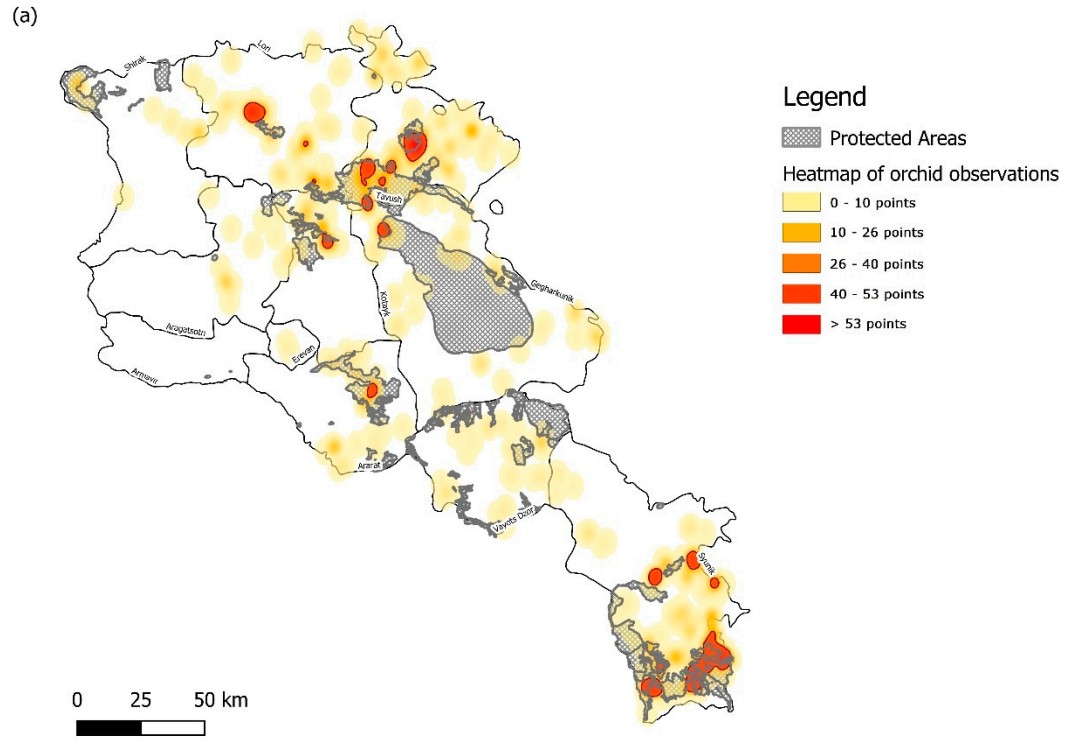

**Figure 1.** *Cont.*

(b)

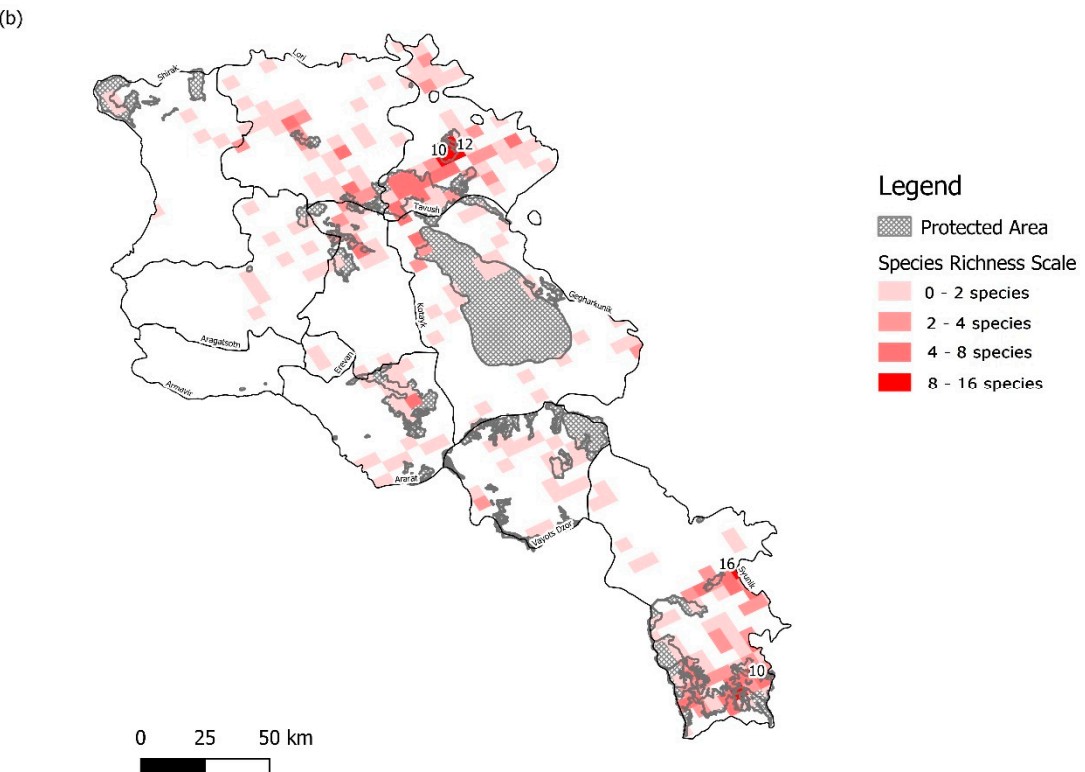

**Figure 1.** Map showing (**a**) a heatmap of orchid observations (genera distribution can be found in Figure S1: Map of genera distribution) and (**b**) species richness within the territory of Armenia, derived from herbarium vouchers, field visits and GBIF records (1101 total data points). Heat map developed using the Kernel Density Estimation with a radial distance buffer of 5 km is graded from yellow to red showing increasing observational findings. Location of Protected Areas derived from the World Database of Protected Areas is also mapped across the study site for comparison.

*3.2. Protected Area Network Comparisons*

Out of the current PA network in Armenia, 57% of the sites held at least one species of orchid (20 out of 35) (Table S2 Species presence absence within PA network in Armenia). The Dilijan National Park, classified as Category II under the IUCN Protected Area Categories System, had the highest orchid diversity (number and evenness) at 2.498 (Table 1). PA sites that were categories under strict protection (Category Ia) had variable richness and diversity, with Shikahogh State Reserve holding 13 species and a diversity index of 2.26, compared to Khosrov Forest State Reserve, holding only six species and a diversity index of 1.50, despite having a larger area (Table 1). Moreover, 14 out of the 43 orchid species (33%) were not found to be represented within any of the protected sites (Table S2 of species presence within PA network in Armenia), including *Steveniella satyrioides* (Spreng.) Schltr., a species listed as EN under the Armenian Red Data Book of Plants [61]. Eight species were only found within one PA site across the entire network, including two species also included within the Armenian Red Data Book of Plants [61], *Cephalanthera kurdica* Bornm. ex Kraenzl. (VU*), from herbarium records and *Corallorhiza trifida* Chatel. (CR) from GBIF (human observation) records. In the case of *C. kurdica,* all known observations for this species were only within the territory of the Arevik National Park.

**Table 1.** Orchid species richness, diversity (Shannon–Weiner Index), area and designation of Protected Area (PA) site found within the territory of Armenia listed by province from north to south. Number of species per PA was calculated using the georeferenced data from herbarium vouchers, field visits and GBIF records. Shannon–Weiner Index was calculated using the "vegan" package [49] in R statistical software [50]. PA names, their reported area and designation are from the World Database of Protected Areas, downloaded from the Protected Planet website [41].

| Protected Area | Orchid Species Richness | Orchid Diversity Index | Reported Area (km²) | Designation | IUCN Category |
|---|---|---|---|---|---|
| Shirak | | | | | |
| *Lake Arpi* | 3 | 0.85 | 212 | National Park | II |
| Lori | | | | | |
| *Gyulagarak* | 4 | 2.41 | 25.8 | Sanctuary | IV |
| *Zikatar* | 1 | 0.00 | 1.50 | Sanctuary | IV |
| *Margahovit* | 4 | 1.36 | 33.7 | Sanctuary | IV |
| Tavush | | | | | |
| *Ijevan* | 14 | 2.41 | 59.1 | Sanctuary | IV |
| *Gandzakar* | 1 | 0.00 | 68.1 | Sanctuary | IV |
| *Dilijan* | 17 | 2.50 | 377 | National Park | II |
| *Akhnabat Yew Grove* | 1 | 0.00 | 0.25 | Sanctuary | IV |
| Gegharkunik | | | | | |
| *Sevan* | 6 | 1.55 | 1474 | National Park | II |
| *Juniper Open Woodland* | 2 | 0.69 | 33.1 | Sanctuary | IV |
| Kotayk | | | | | |
| *Arzakan-Meghradzor* | 4 | 1.21 | 135 | Sanctuary | IV |
| Ararat | | | | | |
| *Khosrov Forest* | 6 | 1.50 | 232 | State Reserve | Ia |
| Vayots Dzor | | | | | |
| *Jermuk Forest* | 1 | 0.00 | 38.7 | Sanctuary | IV |
| *Herher Open Woodland* | 1 | 0.00 | 61.4 | Sanctuary | IV |
| Syunik | | | | | |
| *Arevik* | 17 | 2.30 | 344 | National Park | II |
| *Boghaqar* | 1 | 0.00 | 27.3 | Sanctuary | IV |
| *Shikahogh* | 13 | 2.26 | 121 | State Reserve | Ia |
| *Zangezur* | 2 | 0.69 | 259 | Sanctuary | IV |
| *Goris* | 1 | 0.00 | 18.5 | Sanctuary | IV |
| *Plane Grove* | 1 | 0.00 | 0.64 | Sanctuary | IV |

*3.3. Potential Areas of Suitability and Protection*

Out of the 43 species of orchids found, only 21 had sufficient data points for modeling. PCA analysis of the bioclimatic variables selected showed that 93% of the variation could be explained by the first four Principal Components, with the first (PC1) and second (PC2) Principal Components explaining 77.4% of the overall variation (Figure S2 PCA plot). The maximum temperature of the warmest month (Bio5) and precipitation of the warmest quarter (Bio18) corresponded to PC1 and PC2, respectively, while precipitation of the driest quarter (Bio17) and altitude (Alt) corresponded to PC3 and PC4, respectively (Table S3 Principal Component Analysis results).

Test AUC scores (i.e., the real test of the model's predictive power) for the models varied from relatively low (i.e., model performs close to a random model), in the case of *Epipactis persica* (Test AUC = 0.572), *Platanthera chlorantha* (Custer) Rchb. (0.590) and *Gymnadenia conopsea* (L.) R. Br. (0.597), to those with high predictive power, in the case of *Limodorum abortivum* (L.) Sw. (0.908), *Cephalanthera damasonium* Mill. (0.866) and *Anacamptis pyramidalis* (0.844) (Table 2). Species-specific predicted distributions and response curves can be found in the Supplementary Materials (Figure S3 SDM models per species). Maximum temperature of the warmest month (Bio5) was the best predictor for six of the 21 species, namely for *A. coriophora* (L.) R. M. Bateman, Pridgeon & M. W. Chase and *Orchis*

*punctulata* Steven ex Lindl. Precipitation of the warmest quarter (Bio18) best predicted five species, namely *P. chlorantha* and *Neottia nidus-avis* (L.) Rich., and with regard to *Traunsteinera sphaerica* (M.Bieb.) Schltr., both Bio5 and Bio18 contributed roughly 50/50 to its distribution (Table 2). Precipitation of the driest quarter (Bio17) did not rank highly for any of the species, although for *Dactylorhiza urvilleana* (Steud.) H.Baumann & Künkele, together with Bio5, the variables contributed over 80% of the model's variation, with similar sigmoidal response curves (Figure S3 Species-specific distribution models from MaxEnt). Altitude appeared to be the best predictor for 10 out of the 21 species, although this variable contributed the least within the PCA (Table 2).

**Table 2.** Results from MaxEnt species distribution models for orchids in Armenia. The presence records are the number of points used to build each model, while 25% of the data were set aside to be used to test the model (Test records). The features (*l* = linear, *q* = quadratic, *h* = hinge) and the regularization multiplier used for the best fit model per species are shown. Best-fit model was chosen based on comparing the AICc values of models with regularization multiplier from 1 to 10 for each species (see Table S4: AICc values for model selection using regularization for results). Training and test AUC, with corresponding standard deviation (SD), are also presented, along with the estimate of contributions of the environmental variables (normalized to percentage). Environmental variables were chosen based on a Principal Component Analysis, with max temperature of warmest month (Bio5) corresponding to Principal Component 1 (PC1), precipitation of warmest quarter (Bio18) corresponding to PC2, precipitation of driest quarter (Bio17) corresponding to PC3 and altitude (Alt) corresponding to PC4.

| Species | Presence Records | Test Records | Features | Regularization Multiplier | AICc Value | Training AUC | Test AUC | SD | Variable Contribution (%) | | | |
|---|---|---|---|---|---|---|---|---|---|---|---|---|
| | | | | | | | | | Bio5 | Bio18 | Bio17 | Alt |
| *Anacamptis coriophora* | 24 | 6 | hlq | 1 | 477 | 0.720 | 0.612 | 0.084 | 98.5 | 0.1 | 1.3 | 0.1 |
| *Anacamptis pyramidalis* | 28 | 7 | hlq | 5 | 482 | 0.716 | 0.844 | 0.054 | 0.0 | 0.0 | 0.0 | 100 |
| *Cephalanthera damasonium* | 16 | 4 | hlq | 2 | 285 | 0.860 | 0.866 | 0.050 | 0.0 | 35.3 | 0.0 | 64.6 |
| *Cephalanthera longifolia* | 11 | 3 | lq | 1 | 197 | 0.819 | 0.783 | 0.024 | 0.0 | 18.5 | 0.0 | 81.5 |
| *Cephalanthera rubra* | 33 | 8 | hlq | 3 | 601 | 0.834 | 0.779 | 0.060 | 3.5 | 13.1 | 0.0 | 83.5 |
| *Dactylorhiza euxina* | 15 | 4 | hlq | 2 | 299 | 0.686 | 0.698 | 0.102 | 0.0 | 0.4 | 0.0 | 99.6 |
| *Dactylorhiza incarnata* subsp. *cilicica* | 30 | 8 | hlq | 3 | 574 | 0.795 | 0.710 | 0.084 | 20.5 | 50.2 | 0.0 | 29.4 |
| *Dactylorhiza urvilleana* | 45 | 11 | hlq | 5 | 859 | 0.738 | 0.646 | 0.064 | 56.2 | 13.6 | 30.1 | 0.0 |
| *Dactylorhiza viridis* | 11 | 3 | lq | 1 | 207 | 0.835 | 0.740 | 0.173 | 85.3 | 8.8 | 5.9 | 0.0 |
| *Epipactis helleborine* | 14 | 4 | lq | 1 | 256 | 0.810 | 0.751 | 0.088 | 0.0 | 26.1 | 0.0 | 73.9 |
| *Epipactis persica* | 9 | 2 | l | 3 | 162 | 0.781 | 0.572 | 0.030 | 0.0 | 0.0 | 0.0 | 100 |
| *Gymnadenia conopsea* | 24 | 6 | hlq | 3 | 459 | 0.663 | 0.597 | 0.083 | 0.0 | 0.0 | 0.0 | 100 |
| *Limodorum abortivum* | 11 | 3 | lq | 1 | 192 | 0.857 | 0.908 | 0.017 | 0.0 | 0.0 | 12.2 | 87.8 |
| *Neotinea tridentata* | 15 | 4 | hlq | 2 | 280 | 0.894 | 0.804 | 0.059 | 79.0 | 2.5 | 18.4 | 0.0 |
| *Neottia nidus-avis* | 16 | 4 | hlq | 2 | 270 | 0.782 | 0.799 | 0.083 | 0.0 | 83.5 | 1.6 | 14.9 |
| *Ophrys scolopax* subsp. *cornuta* | 9 | 2 | l | 1 | 169 | 0.834 | 0.839 | 0.011 | 68.7 | 0.0 | 0.1 | 31.2 |
| *Orchis mascula* | 42 | 11 | hlq | 2 | 813 | 0.740 | 0.607 | 0.085 | 20.9 | 40.3 | 10.5 | 28.3 |
| *Orchis punctulata* | 9 | 2 | l | 1 | 170 | 0.853 | 0.660 | 0.087 | 90.8 | 0.0 | 9.2 | 0.0 |
| *Orchis purpurea* | 9 | 2 | l | 2 | 160 | 0.774 | 0.840 | 0.035 | 0.0 | 0.0 | 0.0 | 100 |
| *Platanthera chlorantha* | 23 | 6 | hlq | 3 | 447 | 0.702 | 0.590 | 0.122 | 0.0 | 83.8 | 0.0 | 16.2 |
| *Traunsteinera sphaerica* | 7 | 2 | l | 1 | 135 | 0.770 | 0.766 | 0.108 | 49.8 | 50.2 | 0.0 | 0.0 |

According to the MaxEnt models generated, species that had the highest average range of environmentally suitable area were *Epipactis helleborine* (L.) Crantz (1070 km$^2$), followed by *Neottia nidus-avis* (933 km$^2$) and *Dactylorhiza urvilleana* (793 km$^2$). Those with the narrowest area of suitability area were *Neotinea tridentata* (Scop.) R. M. Bateman, Pridgeon & M. W. Chase (192 km$^2$), followed by *Traunsteinera sphaerica* (221 km$^2$) and *Gymnadenia conopsea* (238 km$^2$) (Figure 2). *N. tridentata* is currently an accepted name for two synonyms of nationally threatened species, *Orchis simia* Vill. (NT) and *O. tridentata* Scop. (EN) (Table S1 Orchid species occurrence data).

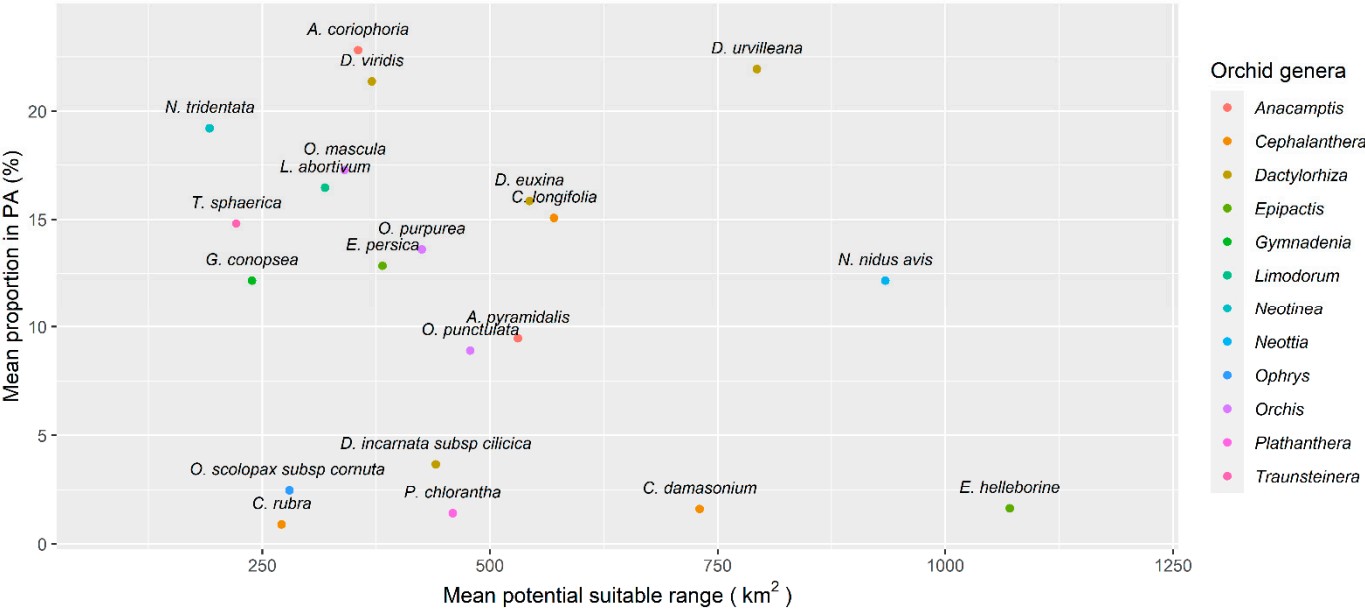

**Figure 2.** Scatterplot showing species potential climatic suitability range (km$^2$) against the mean proportion of species range within a PA represented as a percentage. Colors represent genera according to the World Checklist of Vascular Plants.

We found no significant correlation between the average potential range of suitable area versus proportion of said area under protection (Pearson's correlation: r = −0.211, *p*-value = 0.358). None of the orchid species found had more than 30% of its potential area within the PA network (Figure 2). The two species with the largest potential distribution varied in their levels of protection, with *E. helleborine* having, on average, 1.65% of its range under protection, but *D. urvilleana* showing a higher proportion of its range currently under protection (21.9%) (Figure 2). We found several species with narrow suitability range showing similar protection values to *D. urvilleana*, namely *N. tridentata* (19.2%), *Anacamptis coriophora* (22.8%) and *D. viridis* (L.) R. M. Bateman, Pridgeon & M. W. Chase (21.3%). The study also highlighted species with relatively small suitability ranges that had less than 5% of this under protection, including *Cephalanthera rubra* (L.) Rich. (mean range = 271 km$^2$; mean range protected = 0.9%) and *Ophrys scolopax* subsp. *cornuta* (Steven) E. G. Camus, P. Bergon & A. A. Camus (mean range = 280 km$^2$; mean range protected = 2.48%) (Figure 2), recognized in Armenia as *Ophrys oestrifera* M. Bieb. and a Near Threatened species according to the National Red List (Table S1 Orchid species occurrence data).

## 4. Discussion

Compared to its neighbors, Armenia holds a relatively high number of orchid species for its territorial size. Iran, for example, holds approximately 46 species and subspecies [62], but is more than 14 times larger in land mass than Armenia. The diversity per area of land is comparable to neighboring Georgia, with ~57 species of orchids within a territory that is approximately twice the size [63]. This relatively high diversity of orchids within Armenia, therefore, warrants further protection, both through in situ and ex situ means.

Overall, current PA networks performed well in conserving the orchid species we found, with approximately half of the PA sites capturing orchid species. The distribution pattern of high orchid richness largely focuses on the central and south of the country (Tavush and Syunik provinces), which also correspond to the typical focus for the enhancement of PA networks within the country. After Lake Sevan, protected based on its importance as a key freshwater ecosystem resource, the two largest PAs are within the Syunik and Tavush provinces, respectively [37]. The relatively large size, coupled with PA locality within these provinces, can potentially explain the high orchid richness and diversity captured within. However, over a third of orchids were not represented within any PA site, with a further eight with only one occurrence record within PA. Of particular concern are those recognized as locally threatened, such as *Steveniella satyrioides*. Further uploads onto the WDPA database of newly endorsed PA sites for Armenia are pending, [41]; therefore, it is possible that these new sites will capture the orchid taxa left behind. A clear example is the Khustsup Nature Reserve, which is established, but not yet uploaded into the WDPA system. In addition to enhancing the PA network to include these species, additional conservation measures should be explored, such as ex situ strategies [64], although we recognize that orchids are known as "exceptional species" and may require the use of unconventional seed banking techniques (i.e., cryopreservation and/or in vitro) to ensure long-term conservation of orchid species diversity. Similarly, species only found within one PA and associated with known threatened habitats should also be prioritized for additional studies to fully determine their threatened status within the national context. For example, *Corallorhiza trifida*, recorded in the 1970s and found within forests, could now be lost due to much of the forest being cleared during the economic crises in the 1990s. Therefore, we would prioritize the need to revisit the locality of older records for species with low representation within PAs.

One major issue that we faced was around the taxonomic resolution of orchid species, as several species were recognized as synonyms, thereby reducing the overall species richness and diversity and/or creating uncertainty around conservation status. In our case, two morphologically distinct species, *Orchis simia* Vill., classified as Near Threatened, and *O. tridentata* Scop., classified as Endangered nationally, are both known as synonyms of *Neotinea tridentata*. These taxonomic resolution issues have potential implications within species modeling as the data of two species recognized as synonyms are combined under one globally accepted name, thus potentially overinflating the suitability range for those known locally as narrow endemics, and for *N. tridentata*, the already narrow suitability range found within our study would be of greater concern if this was the case. Difficulties in species-level taxonomy come at a time where taxonomic certainly is increasingly sought after by policy-makers and conservationists. For example, the Global Strategy for Plant Conservation has a target to assess 25% of all plant species, which is largely not reached due to incomplete taxonomic information [65,66]. Complex families, such as Orchidaceae, particularly in plant-rich regions, would need to be prioritized for resolving taxonomically ambiguous taxa. One way that this can be achieved is through integrated taxonomy, which considers both traditional morphological studies and DNA sequence data. Since its inception almost two decades ago, the use of DNA taxonomy has increased substantially with regard to reconstructing phylogenies, particularly for complex families [67,68]. In the Caucasus, there is increasing focus towards the use of DNA within plant-related studies [68]. The region-wide barcoding project currently underway in the Caucasus (CaBOL project: https://ggbc.eu/ (accessed on 28 September 2021)) is considered a first step in capturing DNA data for use in this way, but continued investment in integrated taxonomic studies has the potential to enhance conservation outcomes within biodiverse regions [69–71].

We need to acknowledge the risk of sampling bias with our study, as the occurrence data used were derived predominantly from herbarium records and a publicly accessed georeferenced database (GBIF). As these kinds of data typically do not utilize systematic methodologies, the resulting dataset can be subject to various aspects of sampling bias, such as temporal [72] and geographic bias (e.g., accessibility or known biodiverse areas

such as outskirts of PAs) [73,74]. Using species richness estimates does control for some of these biases; however, we still have the risk of false negatives across our study site (i.e., zero occurrences due to lack of visitation rather than true absence), which is an issue across the study of conservation [73]. Indeed, research continues to discover new populations/localities of orchid species across Armenia [75,76]. Nonetheless, our study did find a potential orchid richness hotspot within the Lori province using our observational data. The Lori province lakes/wetlands have also been highlighted as a key area for plant diversity overall [77]; however, the current PA network within the province is much reduced in comparison with Tavush and Syunik. Therefore, we recommend further investigation within this province to estimate the potential of extending current conservation measures. A more comprehensive recommendation is to enhance PA networks using Key Biodiversity Areas (KBAs), which was described for the Caucasus region in 2020 [37]. There are currently 22 KBA sites within Armenia, with only 3% covered under strict protection (i.e., IUCN Category I) [37]. The areas that we proposed within our study coincide with the current KBAs planned. There is also an opportunity to enhance species protection through incorporating orchid-specific management plans for PAs that are already holding populations of orchids within, as highlighted by our study.

The use of SDMs gives an indication of a species' potential ecological niche and can be used to predict current distribution [78–80]. We found that, overall, the models that were used fitted well, with over 60% showing an AUC value of above 0.7. However, estimating model suitability using AUC has been shown to be problematic, particularly with evidence of sampling bias, and, therefore, model interpretations should be done with caution [81]. Models of three orchid species did show low predictive power, namely *Epipactis persica*, *Platanthera chlorantha* and *Gymnadenia conopsea*. For *E. persica,* model improvement could be achieved through increasing the input of presence data; however, for the other two species, further analysis to incorporate additional environmental variables could be warranted. Nonetheless, our study makes an initial attempt at identifying potential suitability areas for orchids in Armenia, and, in general, our models reflect the known ecology of the target species. For example, *Traunsteinera sphaerica* was shown to be related to areas where precipitation rates remained high, with low maximum temperatures during the warmest months, typical of wetlands/marshes [45]. This close association could also explain their relatively restricted range of suitable areas across Armenia, with highly predicted sites found towards the northern part of the country, where it is relatively humid, with no distinct dry season. Therefore, conservation measures could include increasing the protection and effective management of existing wetland areas and/or restoring degraded marshland areas within the north of the country. The limitation for our model is the exclusion of variables related to habitat, which is commonly used in other modeling studies [82,83]. However, due to the wide temporal range of occurrence data used, some from the 1990s, and the rapidly changing landscape in Armenia due to economic crises, illegal logging and the rise of mining activities, the use of habitat variables could reduce the overall accuracy of predictions. We recognize that further work is needed to refine our models, including increasing field observations for model validation, the incorporation of more refined ecological variables and gathering recent landscape-level changes. Additionally, only a subset of the orchids found in Armenia had enough data points to be suitable for modeling. With increasing threats relating to habitat loss within the region [37], there is an urgent need to accelerate the gathering of locality, ecological and population-related data for these species.

We did not find a significant relationship between species range and proportion under protection, as species with the potential of inhabiting a wider climatic niche had varied protection levels, and vice versa. It is currently uncertain whether additional data for species missing from current SDM analysis will shed further light on a significant relationship between range and protection, which is seen elsewhere [12]. Additionally, the great advantage of SDMs is the ability to predict future distributions and threats regarding

changes in species' potential range, which can assist planning for a sustainable conservation strategy in the face of a changing climate [13,84].

## 5. Conclusions

In conclusion, the establishment and management of PA networks can be a costly process for many nations. Targeting areas where there is high diversity can be an effective strategy to ensure the maximum amount of protection. For Armenia, the current in situ conservation strategies seem to be effective at providing conservation impact for orchids; however, improvements can be made through increasing the proportion of strictly protected sites within already established KBA networks, coupled with enhancing the capacity for ex situ conservation for orchids, particularly those that are found outside of protected sites.

**Supplementary Materials:** The following are available online at https://www.mdpi.com/article/10.3390/d13120624/s1, Figure S1: Map of genera distribution; Figure S2: PCA plot; Figure S3: Species-specific distribution models from MaxEnt; Table S1: Orchid species occurrence data; Table S2: Species presence/absence within PA network in Armenia; Table S3: Principal Component Analysis results; Table S4: AICc values for model selection using regularization.

**Author Contributions:** Conceptualization, A.F. and A.N.; methodology, A.F.; resources, A.P. and A.N.; data curation, A.F., A.N. and A.P.; writing—original draft preparation, A.F.; writing—review and editing, A.N. and A.P.; visualization, A.F.; supervision, A.N.; project administration, A.F.; funding acquisition, A.F. All authors have read and agreed to the published version of the manuscript.

**Funding:** This research was funded by the RBG Kew Pilot Fund.

**Institutional Review Board Statement:** Not applicable.

**Data Availability Statement:** Data will be available upon request to the corresponding author due to the sensitivity of threatened species' locality.

**Acknowledgments:** We would like to express our sincere gratitude to the staff of the herbarium of the Komarov Botanical Institute of the Russian Academy of Sciences, St. Petersburg and personally to L. Averyanov, the Head of LE, and V. Shvanova, a curator of the Caucasian section of LE. We are thankful to our colleague, zoologist Vasil Ananian, for providing us with important photo evidence of orchid species, and to Jenny Williams from RBG Kew for her advice on species distribution and mapping. We also thank volunteer Esther Man for assisting in the curation of the tables for this work. Finally, we thank the five reviewers who took their time to review, comment and make suggestions regarding the manuscript, which allowed us to greatly improve it.

**Conflicts of Interest:** The authors declare no conflict of interest.

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
