# Peer review of "Exploring Effective Conservation of Charismatic Flora: Orchids in Armenia as a Case Study"

_diversity, doi:10.3390/d13120624_

Round 1
Reviewer 1 Report
Exploring effective conservation of charismatic flora: Orchids in Armenia as a case study.
Faruk et al. Reviewer for Diversity
This manuscript presents an interesting study of a charismatic group of plants, in an understudied region of biodiversity importance.
I have some major methodological and interpretation concerns.
- The authors claim to present “potential ranges” of the species but in fact the models they develop should instead be interpreted as climatic niches (since climate variables are the only ones used). Other variables likely to be important in determining species ranges (bedrock, land-cover and land-use, soil pH) are available at global scales and could easily be incorporated in this study, making the models far stronger.
- The authors use MaxEnt to model the distribution of the orchid species. However, the authors do not report on how (or even if) MaxEnt was tuned. Using MaxEnt by just accepting the default settings (for feature classes, regularization etc) is strongly problematic. When MaxEnt is properly tuned for each study species it is a useful approach. For further information see Fourcade, Y., Engler, J.O., Rödder, D., Secondi, J., 2014. Mapping species distributions with MAXENT using a geographically biased sample of presence data: a performance assessment of methods for correcting sampling bias. PLoS ONE 9, e97122. Halvorsen, R., 2013. A strict maximum likelihood explanation of MaxEnt, and some implications for distribution modelling. Sommerfeltia 36, 1–132. Yackulic, C.B., Chandler, R., Zipkin, E.F., Royle, J.A., Nichols, J.D., Campbell Grant, E.H., Veran, S., 2013. Presence-only modelling using MAXENT: when can we trust the inferences Methods Ecol. Evol. 4, 236–243.
- The authors use all 19 WorldClim bioclimate variables to develop the climate niche models. The bioclimate variables are likely to be highly correlated with each other, and including colinear variables in MaxEnt can lead to issues with variance inflation. It is essential to first look at (and present) pairwise correlations between all variables involved in the modelling, and employ variable reduction (e.g. principle coordinate axes, or selection of variables to avoid collinearity) before modelling.
Elevation is likely to also be strongly colinear with bioclimate variables
- The authors fit and report non-significant relationships (Figure 2) which are inappropriate. Once an alpha level (0.05) is chosen, the interpretation must stick to this. The regression line in Figure 2 must be removed, and the reporting of this should be cast as “no relationship” (not a non-significant positive relationship).
Minor comments.
L13: Who criticises the use of PA, and on what grounds?
L15: To “highlight the effectiveness” is a strange way of presenting an objective
L65: An increase in publications on any topic is expected (due to the proliferation of journals and articles). For this to be presented, the number of papers on Orchids should be standardised by e.g. the total number of publications within e.g. plant science (or ecology etc)
L109: It is unclear what data where collected from the sources.
L208: How do these protected area categories relate to the international standards from IUCN?
Figure 1: Increase the shading level of the protected areas, as these cannot easily be seen (or overlay the polygon boundaries ontop of the colours shoing observations
Figure 1: “Species richness” or similar is missing from the label for part b
Discussion: The first paragraph is too tangential. This should focus on your important results and interpretation.
Why not also compare protected areas to non-protected areas?
L348: The use of SDM does not necessarily remove sampling bias- it just translates the bias into the biased sampling of environmental variables
L360: Were occurrence records matched with year of establishment of the protected areas?
Author Response
Dear Reviewer
Thank you for your review, we appreciate the time taken and have carefully considered your comments and suggestions. Please see the attached word file for the changes made to the revised manuscript.
Sincerely,
The Authors

Reviewer 2 Report
The manuscript deals with orchids in Armenia. The topic is interesting and falls within the scope of the journal. The English is ok. The results are well presented and the discussion and conclusions are supported by the results. In the discussion, the authors recognize the limits of their sampling (e.g., line 327 “We need to acknowledge the risk of sampling bias with our study”; line 365 “our results were not statistically significant”). Nevertheless, the conducted study is a step forward the knowledge of Orchidaceae in Armenia.
Therefore, I suggest acceptance after the following minor revisions:
Lines 67-68: I suggest adding some words on pseudocopulation/sexual deception, as follows:
…the curious co-evolution with pollinator systems [18–20], the remarkable cases of sexual deception in some genera like Ophrys L. (Schiestl 2005, Turco et al. 2021), and/or the complex…
Schiestl, FP. On the success of a swindle: Pollination by deception in orchids. Naturwissenschaften 2005, 92(6), 255–264.
Turco, A.; Medagli, P.; Wagensommer, R.P.; D’Emerico, S.; Gennaio, R.; Albano, A. A morphometric study on Ophrys sect. Pseudophrys in Apulia (Italy) and discovery of Ophrys japigiae sp. nov. (Orchidaceae). Plant Biosystems 2021. DOI: 10.1080/11263504.2021.1897702.
Line 72: “Epipactis” instead of “Epipactus”
Line 72: Epipactis in italics
Line 73: Dactylorhiza in italics
Lines 82-83: I suggest adding a more recent reference dealing with the threats affecting orchids with restricted distribution range and the evaluation of their conservation status:
…a greater risk to extinction in the coming decades [23, Wagensommer et al. 2020].
Wagensommer, R.P.; Medagli, P.; Turco, A.; Perrino, E.V. IUCN Red List evaluation of the Orchidaceae endemic to Apulia (Italy) and considerations on the application of the IUCN protocol to rare species. Nature Conservation Research 2020, 5(suppl. 1), 90–101. DOI: 10.24189/ncr.2020.033.
Lines 86-87: “…c.7,000 species of vascular plants, of which 25% are classified…” instead of “…c.7,000 species of vascular plants with 25% are classified…”
Line 112: Add a reference, i.e.: “…Herbarium (LE) (herbaria acronyms according to Thiers 2021)…”
Thiers, B. Index herbariorum. Available from http://sweetgum.nybg.org/science/ih/ (accessed online on 09 Oct 2021)
Line 154: Do not write “(Phillips et al. 2021)”, but “[NUMBER]”. In addition: Write the complete reference of “Phillips et al. 2021” in the reference list (at the moment, it is missing)
Line 159: Do not write “(Swarts and Dixon 2009)” but “[NUMBER]” (maybe number 23 of the reference list)
Line 183: “orchids are Epipactis persica and Epipogium aphyllum” instead of “orchid is Epipactis persica and Epipogium aphyllum”. In addition: Add the authors to the species (in lines 179-180 you wrote the authors of the other species cited the first time in the manuscript)
Line 187: Add the authors to the species “Dactylorhiza fuchsii, D. maculata, Orchis adenocheila and O. calliantha”
Lines 189-190: maybe “occurrences data” instead of “data occurrences”
Figure 1: I suggest indicating the species with different symbols, not all with a point of different colour, that cannot be distinguished at this scale. In addition: Add “a” to the Figure on the left and “b” to the figure on the right
Line 212: “Table 1” instead of “Table 2”
Line 215: “Armenian National Red List”: Add a reference
Line 219: “are only” instead of “is only”
Line 223: “Table 2” instead of “Table 3”
Line 224: Add the authors to D. viridis (first time cited in the text). In addition, write “Dactylorhiza viridis” instead of “D. viridis”
Line 227: “Figure S1” instead of “Figures S3”
Line 229: “three species were found” instead of “three species was found”
Lines 229-230: Add the authors to the species, if cited for the first time in the text
Line 230: “Table 2” instead of “Table 3”
Line 232: “predictor” instead of “predictors”
Line 232: “Table 2” instead of “Table 3”
Line 238: This is Table 1, not Table 2
Line 253: This is Table 2, not Table 3
Line 265: D. viridis in italics
Line 268: “Table 2” instead of “Table 3”
Line 269: Figure 3 does not exist
Lines 265-276: Add the authors to the species, if cited for the first time in the text
Line 285: “Colours represent genera” instead of “Colours represent specific genera”
Author Response
Dear Reviewer 2
We are very thankful for your comments and suggestions that allowed us to greatly improve our manuscript. We have considered all the minor revisions and have take on board all the changes requested.
Please see the attached word file for the changes made to the revised manuscript.
Sincerely,
The Authors

Reviewer 3 Report
The present manuscript is a well-written study, It needs a small correction to be considered for publication. My comments are listed below.
A general comment concerns the quality of English. Although the text is well-readable, I found some difficult sentences or mistakes, Please, double-check the text again in terms of English.
The title, abstract and key words are well-written. The abstract contains the relevant description of the structure and contents of the whole manuscript. Maybe, some numeric data could be added, but it is not a mandatory suggestion.
The section Introduction needs some correction, especially in relation to references. In the first paragraph, saying about biodiversity decline, the authors use "grey literature" and mention a bit small periods (50 years, see line 36), while now there are many interesting suitable papers published in leading journals, such as 1) https://doi.org/10.1016/j.cub.2019.07.063; 2) https://doi.org/10.1007/s10531-020-02000-x; 3) https://dx.doi.org/10.1002/ppp3.10146. I recommend to use them (or, maybe, there are some others) instead of the currently added references [1-3]. Concerning plant extinction, there are some national assessments of the plant extinction level in some large countries, such as https://dx.doi.org/10.1111/cobi.13621, https://dx.doi.org/10.3390/biology10030195, https://doi.org/10.1007/s10531-020-02000-x, https://dx.doi.org/10.3390/d13020078, which could be used to highlight the international relevance of the study of plant extinction issues.
Please, add references after "as these provide a safe refuge for both plants and animals within their natural habitats" in line 42. This important statement should be supported by references, if possible, articles in international journals.
Line 72: Change "Epipactus" to "Epipactis".
I suggest to add a reference for the case study https://dx.doi.org/10.24189/ncr.2020.043 (or other interesting papers of this author) after "the effectiveness of various conservation strategies" in line 72.
In lines 86-100, please, add clear information on how many Protected Areas are existing in Armenia, with classification by types (nature reserves, national parks, etc.).
In line 102, change "highlight" to "reveal". In the Introduction, authors do nothing highlight, while they can plan to do it, but at first, these hotspots should be revealed (demonstrated). And exactly after this action, authors can highlight analyze them.
Text in lines 103-106 should be moved into the section Materials and Methods. Please, note all methods, approaches and other similar points should be described in the section of Materials and Methods.
Finally, please, describe clearer the aim and tasks of this research, as well as hypotheses stated before the study.
The section Materials and Methods needs to be improved, too.
At first, please, explain why did you use LE herbarium, while you didn't investigate MW herbarium? I am especially wondering because MW herbarium is available online, as well as other smaller herbaria. I strongly advise to pay attention to this collection! Of course, this will require re-calculation of data, but I hope that you also aim to obtain unbiased knowledge about the research topic.
Lines 130-133, please, explain to me, did you change the shape-file downloaded from WDPA? I ask you because the WDPA has a significant lack of representativeness of existing protected areas in certain countries. In other words, if you didn't check this file and didn't add existing by missing protected areas, the obtained results could not be correct. That is why, in the Introduction, I ask you how many Protected Areas exist in Armenia. We may expect a certain difference between the actual number of protected areas in Armenia and the number of them represented in WDPA. This is the important issue causing the obtained results. Therefore, I ask you to pay attention to this issue.
Line 156: what do you mean under "occurrence"? Did you consider as different occurrences the findings of a plant in the same location but in different years? It is well-known, that many threatened plants are being collected in herbarium or occurring every year in the same site. Therefore, it is meaningless to count the number of occurrences of the same species in the same site. Such long-term occurrences should be considered as one occurrence made during a period of time (such as 2011-2021).
Lines 157-159: It is wrong to consider that only pollination specificity and mycorrhizal association influence orchid distribution across the landscapes. Please, use the Scopus search to find recently published studies, which demonstrate that many other factors (such as weather conditions, altitude [which was mentioned just as one correlated with the mentioned factors], etc.) play important role in the spatial distribution of orchids. Now, it is a once-side viewpoint.
I have comments on the section Results, too.
Line 108: do you mean that the last orchid collected in Armenia, was found in 2001?? Please, can you explain, whether botanical studies were absent in Armenia during this time? Maybe, the collection of orchids was forbidden? I would be glad to hear the reasons.
I recommend to re-arrange Figure 1 by placing Fig.1A under (not at left) Fig.1B. In such case, they will be wider, and become more readable, because now I cannot understand symbols for certain genera of orchids, because they are too small.
From the sub-section "3.2. Protected Area Network comparisons" I still didn't understand how many and what types of Protected Areas are in Armenia? The mentioning like "e.g. "Sanctuaries" " is meaningless. Why not "national park"? I request again to provide such information because it considerably influences the final results.
Line 215: Add reference after "Armenian National Red List". Where the Red List was published? I would like to read it.
Line 217: Again, add the reference after "National Red List". is it the same as that "Armenian National Red List"?
Please, describe, how the mean range was calculated? Is it related to the geographic (the whole, or within Armenia) range? And a bit related question: how the potential distribution was calculated and how it was applied to the Protected Areas?
In Fig.2, I would suggest enlarging words and symbols for orchids.
The section Discussion is relatively well-written. But it needs to be improved.
For instance, it is great that the orchid diversity was evaluated in Armenia. But it was compared only with Georgia... I strongly suggest comparing your data with orchid diversity in adjacent countries, such as Kazakhstan, Russia, and others. Using Google Scholar, search I found some related research, such as https://dx.doi.org/10.24189/ncr.2021.032 + http://herba.msu.ru/shipunov/school/books/abdulina1999_spisok_sosud_rast_kazakstana.djvu (for Kazakhstan); https://dx.doi.org/10.24189/ncr.2020.018 (for Russia); Shahsavari A (2008) Flora of Iran. Part 57: Orchidaceae. Research Institute of Forests and Rangelands, Tehran (for Iran); C. A. J. Kreutz & A. H. Çolak 2009: Türkiye Orkideleri. — İstanbul: Rota Yayınları + https://doi.org/10.1127/phyto/2019/0292 (for Turkey); and others. Other publications, I suppose, are known to the authors of this paper. This comparison will allow showing the high diversity of orchids in Armenia despite the lower area of its territory.
The text in lines 297-307 is not related to the results obtained during this study. Please, be focused on the discussion of your results in this section without discussing points, which were not targets of the present research.
Lines 308-326 (plus, partially, 327-347): I am repeating again here that if data of the WDPA source were not supplemented by actual data, the authors cannot say about any (non-)representativeness of the Protected Areas network if some PAs were not taken into account.
Concerning SDM analysis, I would advise to include some natural (e.g. landscape-based) parameters to explain the orchid distribution in the future. This is often used in SDM analysis in relation to animals, which provides reliable results.
In general, in the Discussion, some results presented in the section Results, are stated again. Please, try to avoid such repetitions.
In general, the Conclusions section repeats the results presented above in appropriate sections of the manuscript. As it is stated in the Guidelines for Authors, this section should present the main conclusions and implications based on the obtained results and their discussion. Therefore, I kindly ask the authors to improve this section.
Finally, I wish success to the authors during the revision of the manuscript.
Author Response
Dear Reviewer 3
Thank you for taking time to review our manuscript. We have carefully considered the points made. Please see the attached document outlining the changes made based on your recommendations.
Sincerely
The Authors

Reviewer 4 Report
The main problem of this very interesting article, which the authors tried to explain, was to explore whether the current establishment of protected areas is effective at conserving orchid diversity.
The authors used data on orchids distribution in the territory of Armenia, collected from the Herbarium (ERE) of the Institute of Botany NAS RA and the LE Herbarium, as well as field research and GBIF records. Based on the results of analyzes performed with the use of Species Distribution Models (SDMs)/MaxEnt methods, the authors tried to identify spatial patterns of orchid diversity hotspots and corresponding protected areas network sites and highlight the effectiveness of protected areas at capturing orchid species richness and diversity. The authors also tried to investigate the relationship between the size of the species range and the level of protection.
This work is well-structured, well-illustrated and easy to understand. It also addresses a subject that is of great interest in the scientific community. The title clearly describes the contents of the paper. Please consider modifying the abstract by highlighting the components of the work performed. The introduction is well written as it gives a good background of the research in question. I believe that the Materials and Methods section is well structured and scientifically sound. The results are well presented. Literature reviews in the discussion section of the manuscript are good.
The authors are advised to address the following comments for improving the quality of the article.
Lines 72, 73: Please change Epipactus to Epipactis, the names of orchids genera Epipactis and Dactylorhiza should be in italics
Line 86: Please separate with a space: c.(space)7,000
Line 183: Please complete the author of the names, it should be Epipactis persica (Soó) Hausskn. ex Nannf. and Epipogium aphyllum Sw.
Line 187: The names of orchid species used for the first time, i.e. Dactylorhiza fuchsii, D. maculata, Orchis adenocheila and O. calliantha, should include the abbreviation of the author(s) of the name. Please check it through the manuscript.
Line 230: Please separate with a space: D.(space)urvilleana
Lines 262-267 and 271: The abbreviation “subsp.” we write without italics. Please check it through the manuscript
Line 280: Please separate with a space: 162.90(space)km2
Figure 1: In the legend, the names of orchids genera should be in italics
Figure 2: As above, the names of orchids genera (on the right-hand side of the figure) should be italicized
Author Response
Dear Reviewer 4,
We thank you for reviewing our paper and the positive comments made. We have acknowledged the suggested changes to the manuscript (please see the attached document for more details).
Sincerely,
The Authors

Reviewer 5 Report
The analysis of orchid distribution in Armenia is an important addition to our knowledge as this part of Eurasia is not so well depicted.
Author Response
The analysis of orchid distribution in Armenia is an important addition to our knowledge as this part of Eurasia is not so well depicted.
We thank you for your comment, indeed we agree more investigation needs to be done in this region.
Round 2
Reviewer 1 Report
In my opinion the authors have carried out a superficial revision in response to some of my major concerns.
There are two absolutely essential revisions that must me made. These revisions will require re-running the MaxEnt models, with two substantial changes: Firstly due to the strong correlation between the bioclimate variables (as was suspected in the first version, and confirmed by the PCA presented in the revision), one variable representing each principle axis must be selected and the rest ommitted. Otherwise the variance inflation will be servere. An alternative approach would be to use the principle component axes themselves as the environmental variables in MaxEnt.
Secondly, as I suggested in my first review, MaxEnt must be properly tuned to be of any value in interpreting the output. This means that the regularization parameter, and choice of features (linear, quadratic, product, hinge etc) must be selected to fit the data for each species. Without this step, the analysis is inappropriate, and no conclusions can be reliably drawn.
Without these changes, I am unable to support the publication of this manuscript.
Reviewer 3 Report
Dear Authors,
Thank you for considering my recommendations and correcting the manuscript. It became better now. I recommend the manuscript for publication.
Author Response
Many thanks for your comment, it is very much appreciated. Wishing you well.
Sincerely,
The authors